# Romantic relationship configurations and their correlates among LGBTQ+ persons: A latent class analysis

**Fabio Cannas Aghedu[1], Martin Blais [2,3]\*, Léa J. Séguin[3], Isabel Côté[4]**

**1** Team ENACT, University of Nîmes, Nîmes, France, **2** Research Chair in Sexual Diversity and Gender Plurality, Université du Québec à Montréal, Montréal, Québec, Canada, **3** Département de Sexologie, Université du Québec à Montréal, Montréal, Québec, Canada, **4** Département de Travail Social, Université du Québec en Outaouais, Gatineau, Québec, Canada

\* blais.martin@uqam.ca

## Abstract

Research comparing monogamous and non-monogamous relationships on well-being indicators across diverse populations have yielded inconsistent findings. The present study investigates sociodemographic characteristics, as well as personal and relational outcomes, across different relationship configurations. Data were drawn from an online community-based sample of 1,528 LGBTQ+ persons aged 18 years and older in Quebec, Canada. A latent class analysis was performed based on legal relationship status, relationship agreement, cohabitation status, and the seeking of extradyadic sexual and romantic partners on the internet. Class differences on sociodemographic characteristics and well-being and relationship quality indicators were examined. A five-class solution best fit the data, highlighting five distinct relationship configurations: Formalized monogamy (59%), Free monogamy (20%), Formalized open relationship (11%), Monogamous considering alternatives (7%) and Free consensual non-monogamies (3%). Cisgender women were more likely to engage in monogamous relationships than cisgender men, who were overrepresented in open relationships. Lower levels of perceived partner support were observed in both free monogamous and consensually non-monogamous relationships, the latter of which also showed lower levels of well-being. Consensual non-monogamy researchers exploring relationship outcomes should examine relationship facets that go beyond relationship structure or agreement. Variations in monogamies and non-monogamies, both consensual and non-consensual, may be present within each broad relationship configuration, as reflected in different personal and relational needs, which can then translate to better or poorer outcomes.

## Introduction

While monogamy remains the dominant, ideal type of romantic relationship in Western cultures [1, 2], recent research has found that interest in non-monogamy, notably in polyamory, is on the rise [3, 4]. In response to this growing interest, recent papers have delved deeper into

**Data Availability Statement:** Since the data contain potentially sensitive information about study participants, the Université du Québec à Montréal' Human Research Ethics Board has only approved storage of the dataset on secure

institutional servers. Any requests to access the data can be made to Université du Québec à Montréal' Human Research Ethics Board (ciereh@uqam.ca; reference Ethics Protocol Number #2020-2218).

**Funding:** This research is supported by funding from the Social Sciences and Humanities Research Council of Canada (www.sshrc-crsh.gc.ca) awarded to MB and IC (#895-2016-1006). The funders had no role in study design, data collection and analysis, decision to publish, or preparation of the manuscript.

**Competing interests:** The authors have declared that no competing interests exist.

this phenomenon, allowing for more extensive examination and discussion of non-monogamous relationship dynamics and experiences [5–7]. These relationship models are based on the explicit agreement that both partners are free to engage in sexual activity [e.g., open relationships; 4, 8] or in romantic relationships with others [e.g., polyamory; 9, 10]. Recent research in North America suggests that almost 5% of people are currently involved in a non-monogamous relationship [3, 11].

Overall, research suggests that LGBTQ+ persons are more likely to engage in consensually non-monogamous relationships than their heterosexual cisgender counterparts [12–14]. Research has also shown that, among LGBTQ+ individuals, bisexual people are more likely than gay and lesbian individuals to practice non-monogamy [for a review, see 15]. Additionally, recent research has shown that transgender and nonbinary individuals are more likely to be involved in polyamorous relationships than their cisgender counterparts [9, 16, 17]. In a recent U.S. study, 44.6% of bisexual men and 35.0% of bisexual women, compared to 31.6% of gay men and 21.3% of lesbian women reported prior involvement in consensual non-monogamy [3]. According to the same study, 24.6% of heterosexual men and 16.3% of heterosexual women had been in a consensually non-monogamous relationship at some point in their lives. These group differences might be attributable to the fact that, because LGBTQ+ persons are less likely to endorse heteronormative scripts, they are more inclined to consider and explore alternative scripts and relationship configurations [18]. As for bisexual individuals more specifically, Me Lean posits that their greater likelihood of engaging in consensual non-monogamy might be due to the possibility that it allows for the pursuit of sexual and romantic experiences with partners of different genders [2]. Also, another study suggests that the exploration of non-monogamy in and of itself can be an incentive for the exploration of plurisexuality [19].

Because non-monogamous relationship configurations have long been regarded as harmful to both society and individuals [4, 20], researchers have investigated their potential effects on individuals' psychological and relational well-being [21, 22]. Earlier studies have shown that being in a non-monogamous relationship is associated with poor well-being [23, 24]. By contrast, several recent studies have found that consensually non-monogamous relationships present high levels of sexual satisfaction [1], intimacy, commitment, and relationship satisfaction [25, 26]. Moreover, recent research has shown no differences between non-monogamous and monogamous individuals in terms of psychological and relational well-being [22, 27, 28], life satisfaction [21], and health [29]. It is noteworthy that the studies suggesting negative implications of non-monogamous relationships predominantly stem from research conducted in the 1970s and 1980s, while more contemporary investigations tend to emphasize the positive aspects of such relationships. One of the salient factors contributing to these disparities may be the shifting societal perspectives and attitudes toward non-monogamous relationships over time. Another possible explanation for these conflicting findings is the oversight or neglect of potentially confounding variables that may also impact relationship well-being and satisfaction, such as the legal relationship recognition, and infidelity.

Indeed, legal recognition has been documented as an important correlate of relational happiness and well-being [30, 31]. A large number of studies has documented that married couples report greater levels of overall health, happiness, and well-being than unmarried same-gender [32, 33] and different-gender couples [34–37]. It is possible that the observed differences between monogamous and non-monogamous couples could be attributable to legal recognition. However, the association between marriage and positive relational outcomes may not only be due to civil recognition, but also to what marriage entails, such as the division of domestic labor and childrearing duties, the allocation of family resources, and daily social support, all of which are also relevant to unmarried cohabiting couples [30, 31].

Given that most studies have compared married and unmarried people without accounting for cohabitation, it is difficult to determine whether the documented positive relational outcomes are attributable to relationship status (i.e., being married) or to cohabitation more specifically [30]. Yet, studies having compared married and unmarried cohabiting couples have yielded conflicting results. On the one hand, some studies report that marriage, compared to nonmarital cohabitation, is associated with greater well-being [31, 38, 39] and relationship quality among different-sex couples [40, 41]. On the other hand, other studies did not find any differences between cohabitation and marriage [30, 42] or have found that differences in relational well-being favoring married individuals were attributable to other variables, such as gender roles and social and institutional support for marriage [43, 44].

Infidelity has also been found to be associated with lower relationship quality and psychological well-being [45–47]. Recent studies have found infidelity to be strongly associated with insecure attachment [48], relationship dissatisfaction [49], and poor mental health outcomes [47, 50]. Given that approximately 20 to 25% of people in the U.S. [51] report having engaged in extradyadic sexual activity without their partner's knowledge or consent, infidelity is an important factor to consider when examining well-being and relationship quality. However, most research typically defines infidelity as engaging in sexual or romantic activities outside of the primary relationship, without considering the importance of relationship agreement [52, 53]. Consequently, many participants in consensually non-monogamous relationships who reported extradyadic sexual or emotional intimacy could be inaccurately labeled as unfaithful in these studies, potentially leading to biased findings. In this context, it is crucial to emphasize that relationship agreement serves as an essential framework for comprehensively understanding the concept of infidelity and its implications within various relational dynamics.

The internet, including dating applications, has also been known to facilitate extradyadic connections and encounters [54–57]. According to U.S. data, 42% of people using dating applications are in a committed relationship [55]. The seeking of sexual or romantic connections online while being partnered could be examined as an indicator of infidelity not only in monogamous relationships, but also in consensually non-monogamous relationships, depending on the specific relationship agreement (e.g., open relationship; polyamory; etc.).

Most studies having examined diverse relationship configurations concurrently with the aforementioned relationship elements have been conducted among gay men [9]. For this study we have selected five relationship indicators available in our dataset. To our knowledge, no study has investigated legal relationship status, cohabitation, relationship agreement, and the use of internet to seek sexual and romantic partner and their association with psychological and relational well-being in a sexually diverse sample. The first aim of the present study was to address this gap and to identify subgroups characterised by similar multidimensional romantic relationship patterns in a large, sexually diverse sample, using a person-centred approach. In addition, some authors have speculated that certain sociodemographic variables, such as sexual orientation, and socioeconomic status can predict one's propensity to engage in consensually non-monogamous relationships [e.g., 6, 58, 59]. For instance, studies have shown that individuals with higher socioeconomic status are overrepresented in non-monogamous relationships [58, 59]. This overrepresentation may be attributed to financial barriers associated with non-monogamy (e.g., additional costs for travel, activities, and entertainment when dating multiple partners), which can disproportionately impact individuals from marginalized socioeconomic backgrounds.

However, very few studies have assessed the prevalence of these sociodemographic characteristics across relationship configurations. Therefore, the second objective of this study was to explore the relative distribution of these characteristics across various relationship structures.

The present study's third objective was to investigate well-being indicators and perceived partner support across different relationship profiles.

## Materials and methods

### Participants

A community sample (N = 4,746) completed an online survey as part of the Understanding the Inclusion and Exclusion of LGBTQ People (UNIE-LGBTQ) research project. Eligible participants needed to be able to understand French or English, be at least 18 years old, identify as LGBTQ+, and reside in Quebec (Canada). Participants were recruited via printed media, community partners' listservs, websites, and social media, and through word of mouth. To address concerns about illegitimate responses and duplicate entries, we implemented several strategies to verify the authenticity of participant data. These strategies included examining the consistency of responses across survey sections, cross-referencing email and IP addresses to identify potential duplicates, and conducting manual reviews of unexpected responses to identify any patterns indicative of illegitimate responses. People who did not provide any information on their relationship status or duration and who were not currently in a relationship were excluded from analyses (n = 2,037). Further, as some of the selected measures used a 12-month timeframe, participants who had been in a relationship for less than one year were also excluded (n = 367). The analytical sample (n = 1,528) only included participants who provided complete data on the five latent profile indicators (excluding 814 more cases). See Table 1 for the sample's demographic characteristics.

### Measures

**Demographics.** We collected data on respondents' age, education (less than college degree; college education or more), household income, partners' gender (same gender; different gender; partners from multiple genders), and relationship duration. Gender modality (cisgender vs transgender) and gender identity (man, woman, nonbinary) were documented using the Multidimensional Sex/Gender Measure [60]. For analytical purposes, participants were categorized as cisgender men, cisgender women, transgender men, transgender women, and nonbinary individuals. Polyamorous individuals were instructed to respond with their longest relationship in mind. This decision was made to simplify the data collection process and minimize response time. Yet, it may overlook the uniqueness of concurrent relationships, such as differences in investment, satisfaction, commitment, and communication between primary and secondary partners, which are critical to understanding the full spectrum of polyamorous experiences and relationships [61–63].

**Relationship indicators.** *Legal relationship status*. Participants' legal relationship status was assessed using the following question: "What is your legal marital status right now?". Participants were grouped into two categories: those who were in a legally recognized relationship (common law/civil union/married), and those who were not.

*Relationship agreement*. Relationship agreement was evaluated with the question: "What is your current relationship status?", with three options to choose from: (1) monogamous (i.e., having one romantic partner and a monogamous sexual agreement), (2) sexually non-monogamous (having one romantic partner, but a non-monogamous sexual agreement) and (3) polyamorous (having more than one romantic partner).

*Cohabitation status*. Participants were asked: "Do you live with your partner?" (yes/no). Those who were in a polyamorous relationship were asked whether they lived with at least one of their partners.

**Table 1. Sample characteristics.**

| Characteristic | N | % |
|---|---|---|
| **Age** | | |
| 18–24 | 253 | 16.56 |
| 25–29 | 293 | 19.18 |
| 30–39 | 478 | 31.28 |
| 40–54 | 276 | 18.06 |
| 55+ | 228 | 14.92 |
| **Gender of partner(s)** | | |
| Different gender | 291 | 19.94 |
| Same gender | 936 | 61.26 |
| Multiple genders | 24 | 1.57 |
| Missing | 277 | 18.13 |
| **Gender modality and identity** | | |
| Cisgender | 1338 | 87.56 |
| Men | 627 | 41.03 |
| Women | 711 | 46.53 |
| Transgender | 190 | 12.44 |
| Men | 28 | 1.83 |
| Women | 22 | 1.44 |
| Nonbinary | 140 | 9.16 |
| **Education** | | |
| < College degree | 542 | 35.47 |
| College/University degree | 982 | 64.27 |
| Missing | 4 | 0.26 |
| **Household income** | | |
| < $30,000 | 210 | 13.74 |
| $30,000-$59,999 | 305 | 19.96 |
| $60,000-$99,999 | 410 | 26.83 |
| > $99,999 | 515 | 33.70 |
| Missing | 88 | 5.76 |
| **Relationship duration in years** | | |
| 1–5 | 778 | 50.92 |
| 6–10 | 350 | 22.91 |
| 11+ | 400 | 26.18 |

*Internet use to find sexual and/or romantic partners*. Participants were asked whether they sought outside sexual partners ("In the past 12 months, how often have you used the Internet to find a sexual partner?") and outside romantic partners ("In the past 12 months, how often have you used the Internet to find a romantic partner?") on the Internet. Response anchors were: 1 –*Never or almost never*, 2 –*Once or a few times in the last year*, 3 –*Once or a few times a month*, 4 –*Once or a few times a week*, and 5 –*Every day, or almost every day*. For each item, participants were categorized into one of three groups: those who have never or almost never sought a partner on the internet (never or almost never), those who have done so once or a few times in the last year, and those who have done so once per month or more.

**Outcome variables.**   Well-being was measured using the Mental Health Continuum–Short Form [MHC–SF; see 64], a 14-item questionnaire composed of three subscales: (1) emotional well-being (e.g. "How often did you feel interested in life?"), (2) social well-being (e.g.

"How often did you feel that you had something important to contribute to society?"), and (3) psychological well-being (e.g. "How often did you feel that your life has a purpose or meaning?"). Participants were asked to answer each item with the last month in mind. Response options ranged from 0 (*never*) to 5 (*every day*). All three subscales' internal consistency was adequate (Cronbach's α ranging from .83 to .89).

Social provisions from a relationship partner was measured using the 10-item Canadian version of the Social Provisions Scale [SPS; 65]. The scale assesses five two-item constructs: (1) Attachment (e.g., "My romantic partner(s) provide(s) me with a sense of emotional security and well-being"), (2) Reliable alliance (e.g., "I can count on my romantic partner(s) in an emergency"), (3) Guidance (e.g., "I can talk to my romantic partner(s) about important decisions in my life"), (4) Reassurance of worth (e.g., "My competence and skill are recognized by my romantic partner(s)"), and (5) Social integration (e.g., "My romantic partner(s) enjoy(s) the same social activities I do"). Responses ranged from 1 (*strongly disagree*) to 4 (*strongly agree*). Polychoric correlation coefficients showed satisfactory internal consistency for all scales: Attachment ($r = .86$), Reliability ($r = .91$), Guidance ($r = .81$), Reassurance of worth ($r = .89$) and Social integration ($r = .67$).

## Procedure

A link to the survey hosted on LimeSurvey was shared online (via email, listservs, and the project's website and social media), via printed media, and through word of mouth. Once on the LimeSurvey website and before beginning the survey, interested participants were presented a consent form. Upon indicating consent electronically, participants were required to answer eligibility questions. Eligible participants then accessed the survey, which took 50 to 75 minutes to complete. Participants did not receive any incentive for completing the survey. This study was approved by the Institutional Research Ethics Board of the Université du Québec à Montréal (Québec, Canada) (Protocol #2775).

## Data analyses

To identify relationship structure classes, we performed a latent class analysis (LCA) using Latent Gold 6.0 [66]. LCA uses observed indicators to identify homogeneous patterns, or "classes", of specific latent constructs [67]. Five indicators were included to describe relationship structure: (1) social and legal relationship status, (2) cohabitation status, (3) relationship agreement, (4) the seeking of outside sexual partners, and (5) the seeking of outside romantic partners. Latent Gold accounts for missing data using the full information maximum likelihood estimation [68]. First, we estimated LCA models including 1 to 7 classes. The models were compared across fit indices and class sizes. Low Bayesian Information Criterion (BIC), low maximum bivariate residuals, and $L^2$ $p$-values greater than .05 suggest better model fit and entropy values closer to 1 indicate better class separation [69]. Class interpretability was also considered [70]. Second, based on posterior probabilities, participants were assigned to latent classes [71]. Third, to assess class membership differences on sociodemographic covariates, we performed between-class ANOVA-type comparisons implemented in Latent Gold using the maximum likelihood estimator, controlling for age, where the category-specific effects are interpreted in terms of deviation from the average [72]. Class membership differences in outcomes (well-being, and social provision) were explored through multinomial regressions and adjusting for sociodemographic variables such as for age, gender modality and identity, education, gender of the partner, and relationship duration. Class membership differences were estimated using the Bolck-Croon-Hagenaars (BCH) modified bias-correction method, which accounts for uncertainty in class membership [73, 74]. The modified BCH approach has been

recommended for both continuous and binary outcomes variables [73]. At each step, we computed robust variance estimations to prevent standard errors underestimation. When omnibus tests utilizing Wald's criterion revealed significant between-class differences, we examined the Bonferroni-corrected *post hoc* comparisons to maintain the familywise Type I error rate at 0.05 (by dividing the Type 1 error rate by the number of pairwise comparisons). Missing data were handled via Latent Gold default parameters [72].

## Results

### Sample characteristics

On average, participants were 37.03 years old ($SD$ = 13.57). The analytical sample was relatively evenly distributed across age groups (see Table 1). Approximately 87% of respondents were cisgender, and 13% were transgender or nonbinary. Over half of participants reported holding a college or university degree (64%), and 85% indicated having a household income greater than $30,000 CAD. Approximately half of respondents reported a relationship duration of 1 to 5 years (50%).

### Latent classes

A five-class solution was identified as the optimal model (see Table 2). The indicators' conditional probabilities for each class are presented in Table 3. Class 1 (59%; formalized monogamy) includes participants in legally recognized unions (married, civil union, or common-law) with a monogamous agreement, who are cohabiting, and who had never or rarely used the internet to seek another sexual or romantic partner in the last year. Class 2 (20%; free monogamy) was composed of participants in relationships that were not legally recognized, who had a monogamous agreement, who were not cohabiting, and who had never or almost never used the internet to seek another sexual or romantic partner in the last year. Class 3 (7%; monogamous considering alternatives) included participants who were monogamous and cohabiting and had used the internet once or a few times over the previous year to seek sexual and romantic partners. Class 4 (11%; formalized open relationship) described participants who were in open, cohabiting, and legally recognized unions, and who used the internet once a month or more often to find sexual, but not romantic partners, in the last year. Class 5 (3%; free consensual non-monogamies) was the smallest and included participants who were in open relationships that had not been formalized, and who had used the internet once a month or more often in the past year to look for other sexual and romantic partners.

**Table 2. Goodness of fit indices of LCA models.**

| Number of classes | LL | N of parameters | AIC | BIC | L² *p*-value | Max. BVR | Entropy |
|---|---|---|---|---|---|---|---|
| 1 | -4279.78 | 8 | 8575.57 | 8618.22 | *p* < 0.00 | 553.95 | 1.00 |
| 2 | -3849.46 | 17 | 7732.91 | 7823.55 | *p* < 0.00 | 548.70 | 0.84 |
| 3 | -3625.35 | 26 | 7302.70 | 7441.32 | *p* < 0.00 | 27.29 | 0.81 |
| 4 | -3562.34 | 35 | 7194.67 | 7381.28 | *p* < 0.00 | 21.54 | 0.83 |
| **5** | **-3514.72** | **44** | **7117.45** | **7352.04** | ***p* = 0.01** | **2.79** | **0.82** |
| 6 | -3501.39 | 53 | 7108.79 | 7391.37 | *p* = 0.15 | 2.20 | 0.82 |
| 7 | -3487.15 | 62 | 7098.31 | 7428.88 | *p* = 0.82 | 0.11 | 0.80 |

LL = Log-likelihood; AIC = Akaike Information Criterion; BIC = Bayesian Information Criterion; L² = Likelihood-ratio goodness-of-fit; Max. BVR = maximum bivariate residuals. Bold indicates the model was selected for further analyses.

Table 3. Class description across relationship indicators.

| | Full sample | Class 1 | Class 2 | Class 3 | Class 4 | Class 5 |
| --- | --- | --- | --- | --- | --- | --- |
| | | Formalized monogamy | Free monogamy | Monogamous considering alternatives | Formalizedopen relationship | Free consensual non-monogamies |
| Size (%) | | 58.90 | 19.99 | 6.79 | 11.43 | 2.89 |
| **Relationship agreement (%)** | | | | | | |
| Monogamous | 82.66 | **95.23** | **94.71** | **57.84** | 29.49 | 11.72 |
| Open | 13.87 | 3.73 | 3.5 | 29.11 | **61.41** | **68.59** |
| Polyamorous | 3.47 | 1.04 | 1.79 | 13.05 | 9.1 | 19.69 |
| **Legal recognition (%)** | | | | | | |
| Not Married | 28.21 | 6.18 | **90.86** | **51.47** | 0.44 | **98.80** |
| Common law/civil union/ married | 71.79 | **93.82** | 9.14 | 48.53 | **99.56** | 1.20 |
| **Cohabitation status (%)** | | | | | | |
| Non-cohabitation | 17.15 | 0.19 | **62.98** | 39.02 | 1.11 | **57.61** |
| Cohabitation | 82.85 | **99.81** | 37.02 | 60.98 | **98.89** | 42.39 |
| **Used the Internetto find romantic partners (%)** | | | | | | |
| Never or almost never | 86.13 | **98.47** | **94.5** | 37.07 | **53.78** | 19.93 |
| Once or a few times in the last year | 6.94 | 1.52 | 4.14 | **61.74** | 6.9 | 8.03 |
| Once a month or more often | 6.94 | 0.01 | 1.36 | 1.19 | 39.32 | **72.03** |
| **Used the Internetto find sexual partners (%)** | | | | | | |
| Never or almost never | 76.77 | **95.3** | **95.8** | 14.25 | 4.44 | 0.58 |
| Once or a few times in the last year | 10.01 | 4.42 | 2.19 | **83.18** | 11.55 | 0.11 |
| Once a month or more often | 13.22 | 0.29 | 2.01 | 2.58 | **84.01** | **99.31** |

Note: Percentages in bold represent the most distinctive features of each class.

## Class differences on sociodemographic characteristics

Table 4 shows the five latent classes' sociodemographic composition. Bonferroni-corrected pairwise comparisons revealed that participants in the *free monogamy* class were younger than those in *formalized monogamy* and *formalized open relationship* classes (all $p < .001$). As age is a potential confounder for most sociodemographic variables [75, 76], all subsequent Wald tests and corresponding $p$-values were computed on age-adjusted proportions. The *free monogamy*, *monogamous considering alternatives* and *free consensual non-monogamies* classes reported lower relationship durations than those in other classes (all $p < .001$). The *formalized monogamy* and *formalized open relationship* classes reported higher household incomes ($> \$99,000$ CAD) than the other classes (all $p < .001$).

Bonferroni-corrected between-class comparisons also revealed that cisgender men were overrepresented in the *formalized open relationship* (72%) and *free consensual non-monoga-mies* (68%) classes and underrepresented in the *formalized monogamy* (33%) and *free monog-amy* (38%) classes. Conversely, cisgender women were significantly more likely to belong to the *formalized monogamy* (55%) and *free monogamy* (52%) classes than to any other class (all $p < .001$). While nonbinary individuals predominantly belonged to the *monogamous considering alternatives* and *formalized open relationship* classes, the prevalence of nonbinary persons in the remaining classes were not statistically different from one another, likely due to limited

**Table 4. Sociodemographic composition of latent classes.**

| Sociodemographic variables | Total sample | Class 1: Formalized monogamy (ref) | Class 2: Free monogamy | Class 3: Monogamous considering alternatives | Class 4: Formalized open relationship | Class 5: Free consensual non-monogamies | p-value |
|---|---|---|---|---|---|---|---|
| **Age, M (SE)** | 37.03 (0.34) | 38.16 (0.44) | 32.92 (1.00) | 35.25 (1.46) | 39.33 (0.93) | 37.60 (2.71) | <.001 |
| **Gender modality and identity (%)** | | | | | | | <.001 |
| Cisgender men | 41.04 | 33.02 | 38.55 | 54.12 | 72.19 | 68.13 | |
| Cisgender women | 46.2 | 54.69 | 51.97 | 26.15 | 10.18 | 22.73 | |
| Transgender men | 2.14 | 2.62 | 0.71 | 1.41 | 3.11 | 0 | |
| Transgender women | 1.38 | 1.86 | 0.44 | 1.05 | 1.07 | 0 | |
| Non-binary | 9.24 | 7.81 | 8.32 | 17.26 | 13.45 | 9.14 | |
| **Gender of partner(s) (%)** | | | | | | | <.05 |
| Different gender | 23.32 | 24.4 | 21.3 | 21.9 | 22.45 | 21.94 | |
| Same gender | 74.62 | 75.39 | 78.31 | 69.51 | 67.96 | 71.59 | |
| Multiple genders | 2.06 | 0.21 | 0.39 | 8.59 | 9.59 | 6.47 | |
| **Education (%)** | | | | | | | .24 |
| < College degree | 35.53 | 34.22 | 41.73 | 32.43 | 34.73 | 29.68 | |
| College/University degree | 64.47 | 65.78 | 58.27 | 67.57 | 65.27 | 70.32 | |
| **Household income (%)** | | | | | | | <.001 |
| < $30.000 | 14.2 | 9.26 | 26.74 | 24.55 | 7.49 | 30.54 | |
| $30.000-$59.999 | 21.93 | 18.88 | 31.17 | 24.93 | 16.27 | 35.64 | |
| $60.000-$99.999 | 28.96 | 30.05 | 29.78 | 27.67 | 27.59 | 9.49 | |
| > $99.999 | 34.91 | 41.81 | 12.31 | 22.85 | 48.64 | 24.33 | |
| **Relationship duration (%)** | | | | | | | <.001 |
| 1–5 | 53.18 | 42.26 | 87.55 | 64.32 | 37.62 | 73.7 | |
| 6–10 | 22.37 | 27.2 | 7.63 | 14 | 29.39 | 17.67 | |
| 11+ | 24.45 | 30.55 | 4.82 | 21.68 | 32.99 | 8.63 | |

*Notes.* Excepted for age, all proportions were adjusted for age

statistical power afforded by the small subsample of nonbinary individuals. Regarding partner gender, unadjusted results indicate that individuals in the *monogamous considering alternatives*, *formalized open relationship*, and *free consensual non-monogamies* classes were more likely to be paired with multiple genders (all $p < .05$). However, the Bonferroni-corrected post-hoc comparisons yielded nonsignificant results.

## Class differences on well-being and social provisions from partner

Table 5 shows between-class differences on well-being and social provisions from a relationship partner. After adjusting the models for potential confounders (age, gender modality and identity, education, partner gender, and relationship duration), participants in the *formalized monogamy* class reported higher levels of social and emotional well-being than those in the *formalized open relationship* class. No other class differences on well-being indicators were found. Regarding social provision indicators, individuals in the *free monogamy* class indicated lower levels of reliable alliance, guidance, and social integration compared to those in the *formalized monogamy* class. Similarly, the *free consensual non-monogamies* class reported lower

**Table 5. Class differences on well-being indicators and social provisions from relationship partner.**

| | Class 1 | Class 2 | Class 3 | Class 4 | Class 5 | *p*-value | Differences between classes |
|---|---|---|---|---|---|---|---|
| | Formalized monogamy | Free monogamy | Monogamous considering alternatives | Formalized open relationship | Free consensual non-monogamies | | |
| | M (SE) | M (SE) | M (SE) | M (SE) | M (SE) | | |
| **Well-being** | | | | | | | |
| Emotional | 3.80 (.03) | 3.54 (.07) | 3.58 (.08) | 3.58 (.10) | 3.28 (.17) | <.001 | 1>2***; 1>4*; **1>5**\*\* |
| Psychological | 3.55 (.04) | 3.28 (.07) | 3.41 (.09) | 3.48 (.10) | 3.15 (.17) | =.003 | 1>2*** |
| Social | 2.75 (.04) | 2.47 (.07) | 2.73 (.10) | 2.58 (.13) | 2.17 (.19) | =.002 | 1>2*; **1>5**\* |
| **Social Provisions Scale** | | | | | | | |
| Attachment | 3.79 (.02) | 3.70 (.03) | 3.75 (.04) | 3.65 (.06) | 3.52 (.09) | =.001 | **1>5**\* |
| Reliable alliance | 3.89 (.01) | 3.75 (.03) | 3.81 (.03) | 3.75 (.05) | 3.73 (.07) | <.001 | **1>2**\*\*\* |
| Guidance | 3.84 (.01) | 3.72 (.03) | 3.78 (.04) | 3.76 (.05) | 3.52 (.08) | <.001 | **1>2**\*\*\***;1>5**\*\*\* |
| Reassurance of worth | 3.75 (.02) | 3.64 (.03) | 3.70 (.04) | 3.71 (.05) | 3.54 (.08) | =.003 | 1>2* |
| Social integration | 3.53 (.02) | 3.33 (.04) | 3.43 (.04) | 3.32 (.08) | 3.28 (.10) | <.001 | **1>2**\*\*\* |

*Notes*. Differences between classes in bold are significantly different at p< .05\*, p< .01\*\*, p< .001\*\*\* (Bonferroni corrected) in the two-sided test of Wald for adjusted means equality (adjustment variables: age, gender modality and identity, education, gender of the partner, and relationship duration); M = mean; SE = standard error. All estimates are measurement-error weighted based on the BCH approach in Latent GOLD 6.0.

levels of guidance and attachment from their relationship partners than individuals in the *formalized monogamy* class.

## Discussion

This research used LCA to investigate relationship structure patterns and their correlates among LGBTQ+ individuals in Canada. Five latent classes were identified. Two classes were composed of participants in a monogamous relationship: those who were legally married and cohabiting (formalized monogamy; 59%), and those who were legally single and non-cohabiting (free monogamy; 20%). The three other classes were participants with non-monogamous practices. One class was composed of cohabiting participants with a monogamous relationship agreement who sought outside sexual and/or romantic partners on the internet once or a few times a year (monogamous considering alternatives; 7%). The other two classes were composed of consensual monogamous participants: those in a legally recognized relationship with a non-monogamous relationship agreement who cohabit and who often seek outside sexual partners on the internet (formalized open relationship; 11%), and those who are legally single with a non-monogamous relationship agreement, who do not live with their partner, and who often seek outside sexual and/or romantic partners (free consensual non-monogamies; 3%).

The proportion of participants involved in consensual non-monogamy in the overall sample (14% open relationship; 3% polyamorous) was greater than that reported in previous

studies [3, 10], likely reflecting the absence of cisheterosexual persons in the present sample. As suggested by Rutherford et al., LGBTQ+ persons' non-normative sexuality and gender expression may provide a context that is conducive to the development of non-normative relationship configurations in which sexual and affective exclusivity are negotiated rather than assumed [17].

Sociodemographic differences were found between classes. First, congruently with previous research [8, 11, 25, 77, 78], the present study showed that cisgender sexual minority women were more likely than cisgender sexual minority men to engage in monogamous relationships, and the latter were more likely than the former to be in open relationships. Some studies suggest that lesbian women may have a greater adhesion to heteronormative gender roles prescribing monogamy, marriage, and family [79, 80] than do gay men. This discrepancy may be attributed to apprehensions linked to sexual double-standards and the societal censure that women might encounter when articulating their sexual inclinations and preferences within consensually non-monogamous relationships [52]. In contrast, gay men have been found to be more resistant to heteronormative sexuality, allowing them to more freely embrace diverging relationship ideals and configurations or to resist those that are socially enforced [81]. Some evolutionary psychology hypotheses suggest the existence of sex differences in mating strategies, with cisgender women being more likely to endorse monogamy, and cisgender men, non-monogamy [82, 83].

Contrary to previous findings [58, 59], we did not find white and higher household income participants to be overrepresented in non-monogamous classes. In the present sample, the free monogamy class was comprised of individuals reporting lower household income, which may reflect the overrepresentation of younger participants in this class. The fact that these individuals generally have lower personal incomes and are less likely to cohabitate with their partners can negatively affect their household income.

Further, in the present study, educational attainment was not significantly associated with class membership. Research having examined this question presents conflicting findings, with some studies suggesting lower levels of education among individuals engaged in non-monogamous relationships [84], and others reporting greater proportions of highly educated individuals among non-monogamous participants [59]. These inconsistencies suggest the involvement of other unmeasured variables and underscore the need for further investigation.

The present study also found that participants in the *free consensual non-monogamies* class reported lower levels of emotional and social well-being than those in the formalized monogamy class. Similarly, individuals who were legally single (i.e., *free monogamy* and *free consensual non-monogamies*) reported lower overall levels of perceived partner support than those in the *formalized monogamy* class. Such group differences could be attributable to several factors. First, choosing or intending to have one's relationship legally recognized might reflect preexisting relationship characteristics, such as greater levels of satisfaction [85] or mutual support. Second, between-group differences in perceived partner support may be indicative of relational marginalization, that is, the stigmatization of same-sex unions at the family, community, and societal levels and its detrimental impact on such relationships [86, 87]. While legal recognition can provide some validation to same-sex unions, its absence might lead to further marginalization or stigmatization, which are known to negatively affect relationship functioning [88].

A third possible explanation for the low levels of perceived partner support within the *free consensual non-monogamies* class is the possibility that some individuals in the former group were in a transitional stage toward an open relationship. Many non-monogamous relationships were initially monogamous, and the process of negotiating rules and setting boundaries, as well as the trials and errors that many partners engage in when first practicing non-

monogamy can create a context that fosters relationship instability, which could feed into perceptions of lower partner support.

Finally, it is also possible that some participants in the *free consensual non-monogamies* class reluctantly consented to a non-monogamy agreement or were otherwise dissatisfied with their relationship agreement. One recent study suggest that non-monogamous relationships characterized by lower levels of mutual consent and comfort regarding their relationship agreement demonstrated the lowest levels of individual and relationship functioning [89]. Future non-monogamy research examining well-being and relationship quality should account for relationship stage (i.e., transitional/negotiation stage) as well as consent and comfort vis-à-vis relationship agreement.

The absence of significant group differences on well-being indicators is inconsistent with studies that have documented poorer well-being among individuals in non-monogamous relationships [23, 24]. However, this finding aligns with research showing no differences between monogamous and non-monogamous individuals regarding well-being [22, 27], life satisfaction [21], health, and happiness [29]. Some studies also suggest that compersion—the positive attitudes, thoughts, and actions that arise in response to one's intimate partner's extradyadic intimate relationships [90]—significantly correlates with relationship satisfaction in consensually non-monogamous relationships [90–92]. As Rubel and Bogaert hypothesized, it might not be the adhesion to non-traditional relationships *per se* that predicts negative outcomes, but intrarelational characteristics that go beyond sexual and romantic exclusivity [22].

## Strengths and limitations

To our knowledge, this study is the first to examine different patterns of relationship configurations and to investigate group differences on sociodemographic characteristics, well-being indicators, and perceived partner support in a large LGBTQ+ sample. However, some limitations should be noted. First, its cross-sectional and retrospective design is subject to recall bias, precluding any causal inferences. Further, as all data were self-reported, they may be subject to social desirability bias. Because the sampling for this study was convenient (non-probabilistic), no claim about the representativeness of our results can be made. In addition, considering that multiple and diverse recruitment strategies were utilized, the present findings may not be generalizable to the broader population of LGBTQ+ individuals. Further, relationship agreement was measured differently for monogamous and sexually non-monogamous (i.e., open) relationships than it was for polyamorous relationships. Number of partners (i.e., one romantic partner) and relationship agreement (i.e., having a monogamous/non-monogamous sexual agreement) was assessed in the former, while only the number of partners (i.e., having more than one romantic partner) was assessed in the latter. This means that participants with an open relationship agreement did not need to have any outside sexual partners at the time of the study, while those with a polyamorous agreement needed to be actively practicing polyamory to be categorized as polyamorous. The results might have been different if relationship agreement had been measured the same way across relationship types. Additionally, it is important to note that we only collected participants' relationship agreements, which might differ from what their partners would have answered. Lastly, the measure assessing internet use to seek outside sexual/romantic partners may not have fully captured the extent of extradyadic romantic and sexual involvement and having "Never or almost never" as a response anchor rather than separate "Never" and "Almost never" anchors could have limited the measure's precision, muddling possible distinctions between individuals who are consistently faithful and those who are not.

## Implications

While the current findings highlight the importance of considering different types of relationship configurations in future studies–notably in LGBTQ+ samples–they also suggest that relationship agreement alone is insufficient when examining well-being and relationship quality. Indeed, cohabitation, legal relationship status, and the seeking of outside romantic or sexual partners online, can also influence relational outcomes. To better understand the associations between romantic relationships and individual and relational well-being among LGBTQ+ individuals, future research should consider additional variables such as the intention or desire to formalize one's relationship, relationship secrecy or perceived social support for one's relationship, satisfaction with one's relationship agreement, and internalized sexual stigma. Future infidelity research would also benefit from accounting for relationship agreement or configuration to avoid misclassifying individuals practicing consensual non-monogamy as unfaithful. Future relationship diversity research collecting data from both (or more) partners could also yield greater insight.

Future programs for LGBTQ+ individuals, as well as therapists working with these populations, should not assume monogamy and should account for the diversity of sexual and romantic configurations. Therapists working with individuals in consensually non-monogamous relationships also should not assume that they involve infidelity or that their relationship agreement is the cause of poor relationship functioning. These approaches can foster a more inclusive environment that promotes healthy relationships and overall well-being. Considering the prevalence of seeking outside sexual partners online, sexual health education should also incorporate discussions about safer practices, consent, and risk reduction in online encounters. This education can empower individuals to make informed decisions and mitigate potential health risks associated with extradyadic sexual activities.

## Conclusion

The present study explored various relationship structures among LGBTQ+ individuals in Quebec (Canada), identifying five distinct patterns ranging from formalized monogamy to non-monogamous agreements. The prevalence of non-monogamous relationships in this sample exceeded previous estimates, reflecting the unique sociocultural context of LGBTQ+ individuals. Gender played a significant role, with cisgender women more inclined towards monogamy, while cisgender men leaned towards non-monogamy, challenging conventional assumptions. The study also revealed disparities in perceived partner support across different relationship classes, potentially linked to factors such as the cultural significance of marriage, internalized sexual stigma, and relationship stage (e.g., pre-, post-, or during transition from monogamy to consensual non-monogamy). However, no significant differences were found in well-being between monogamous and non-monogamous individuals, suggesting that relationship dynamics may be a more important factor than relationship agreement per se. This study offers valuable insight into the diverse landscape of LGBTQ+ relationships in Canada, emphasizing the need for nuanced exploration of the factors shaping these relationships, with implications for research and support services within the LGBQ+ community. Understanding this complexity is also essential for creating healthier, more inclusive, and supportive environments for relationship diversity among LGBQ+ communities.

## Acknowledgments

The Understanding the Inclusion and Exclusion of LGBTQ People research (UNIE-LGBTQ) is a research partnership of universities, public agencies, semi-public and community-based organizations, and private enterprises dedicating their efforts to better understand situations

in which LGBTQ + people are demeaned, rejected, and belittled, and deprived of the full extent of their rights in important life domains. The authors extend their gratitude to their research partners as well as to the participants who generously shared their experiences with them.

## Author Contributions

**Conceptualization:** Fabio Cannas Aghedu, Martin Blais.

**Data curation:** Fabio Cannas Aghedu, Martin Blais.

**Formal analysis:** Fabio Cannas Aghedu, Martin Blais.

**Funding acquisition:** Martin Blais, Isabel Côté.

**Investigation:** Martin Blais.

**Methodology:** Fabio Cannas Aghedu, Martin Blais.

**Project administration:** Martin Blais.

**Resources:** Martin Blais.

**Software:** Martin Blais.

**Supervision:** Martin Blais.

**Validation:** Martin Blais, Isabel Côté.

**Visualization:** Fabio Cannas Aghedu, Martin Blais.

**Writing – original draft:** Fabio Cannas Aghedu, Martin Blais.

**Writing – review & editing:** Fabio Cannas Aghedu, Martin Blais, Léa J. Séguin.

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
