## [Decision Letter · Decision Letter 0]

5 Apr 2024

PONE-D-24-03060Romantic relationship configurations and their correlates among cisgender sexual minority persons: A latent class analysisPLOS ONE

Dear Dr. Blais,

Thank you for submitting your manuscript to PLOS ONE. After careful consideration, we feel that it has merit but does not fully meet PLOS ONE’s publication criteria as it currently stands. Therefore, we invite you to submit a revised version of the manuscript that addresses the points raised during the review process.

We look forward to receiving your revised manuscript.

Kind regards,

Ning Cai, Ph.D.

Section Editor

PLOS ONE

Journal Requirements:

**Additional Editor Comments:**

The reviewers critique a manuscript on monogamous arrangements among LGBTQ individuals, acknowledging the manuscript's clarity but questioning its novelty and the clarity of its analyses. They highlight issues with the presentation of results and the justification for using adjusted p-values. Recommendations include performing one-way ANOVAs and providing clear reasons for the inclusion of additional variables. The reviewers also suggest the manuscript should consider biological explanations for gender differences in monogamy preferences and question the aggregation of diverse ethnic groups into a single BIPOC category.

Reviewers' comments:

Reviewer's Responses to Questions

**Comments to the Author**

1. Is the manuscript technically sound, and do the data support the conclusions?

Reviewer #1: Yes

Reviewer #2: Yes

Reviewer #3: Partly

2. Has the statistical analysis been performed appropriately and rigorously? 

Reviewer #1: Yes

Reviewer #2: Yes

Reviewer #3: No

3. Have the authors made all data underlying the findings in their manuscript fully available?

Reviewer #1: Yes

Reviewer #2: No

Reviewer #3: No

4. Is the manuscript presented in an intelligible fashion and written in standard English?

Reviewer #1: Yes

Reviewer #2: Yes

Reviewer #3: Yes

5. Review Comments to the Author

Reviewer #1: The main claim of the paper concerns the need to investigate intra-relational characteristics that go beyond relationship structure or agreement. The claim is significant because highlights the importance of taking into account relationships in their complexity beyond their agreements of (non)exclusivity and not taking monogamy for granted, especially for sexual minority individuals. The analysis supports the claim.

The literature cited is sufficiently extensive, however I would add some other recent contributions for the study of CNM such as the special section in the journal Archives of Sexual Behavior (Hamilton et al., 2021) and the special issue focusing on parenting practices published for Sexualities (Klesse et al., 2022). In reference to the intersection between plurisexual orientations and non-monogamies, other studies have highlighted how the exploration of plurisexuality often runs parallel with the exploration of consensual non-monogamy, or the exploration of non-monogamy is an incentive for the exploration of plurisexuality (and not just viceversa) (see Braida, 2021 in E. Maliepaard & R. Baumgartner (eds.), Bisexuality in Europe).

The manuscript is well organized and written clearly enough to be accessible to non-specialists. I only recommend reviewing the use of parentheses, because several times in the text they are opened and not closed, or there are double parentheses.

The methodology and data analysis performed are explained in sufficient detail. Data are held on secure institutional servers and they can be consulted upon request.

Reviewer #2: Review of the article: Romantic relationship configurations and their correlates among cisgender sexual minority persons: A latent class analysis

The article entitled “Romantic relationship configurations and their correlates among cisgender sexual minority persons: A latent class analysis” presents the results of a latent class analysis (LCA) on a LGBTQ+ sample from Quebec, Canada, depicting the comparison between monogamous and non-monogamous relationships in terms of various characteristics. Data were drawn from an online sample of 1338 cisgender sexual minority participants. LCAs revealed five distinct relationship configurations: Formalized monogamy, Free monogamy, Monogamous considering alternatives, Formalized open relationship, and Free consensual non-monogamous. Further analyses that compared several sociodemographic variables between these classes showed that cisgender women were more likely to engage in monogamous relationships than cisgender men, who were overrepresented in open relationships. Though groups did not differ significantly in most well-being indicators, lower levels of perceived partner support were observed in both free monogamous and free CNM relationships.

The article is clearly written, with a good number of up-to-date articles cited, and presents analyses conducted with accuracy and in accordance with best standards. The selection of variables used to perform LCA is reasonable. The sample size is large and has a great age range. The results are presented clearly and are understandable to both readers familiar with the topic and novices. The discussion is well suited to the results, and Authors discuss each of the results with reference to relevant literature. Overall, I think that the study is a valuable addition to the ever-expanding knowledge of non-monogamy, especially because it presents a quantitative measurement, in contrast to the large number of studies in this field using mainly qualitative methods.

Though, I have a small number of comments to which I’d like Authors to refer:

1. The authors conducted the study on a sample of LGBTQ+ people, but only on the cisgender group. Information about excluding transgender individuals (N = 190) and transferring their sample to another study is provided, but in my opinion, this step requires justification. At no point do the authors argue for reasons for such action, for example, by citing literature that shows that transgender individuals differ significantly from cisgender individuals in terms of relationships characteristics and other variables included in the study (if such literature exists). It would probably be valuable to also include this sample in this study and compare the results to those of the cisgender group. Otherwise, more justification of the exclusion of transgender individuals should be made.

2. Authors stated that “Polyamorous individuals were instructed to respond with their longest relationship in mind”. It is a common problem, namely, how to conduct research on CNM people to measure across all their partners. It may be problematic not only due to the complexity of such measurement, but also because biased responses by CNM individuals who could respond according to the anchoring heuristic or average their ratings, which would not provide adequate results. This comment is in no way intended to be a criticism of the authors, and I believe they made the right decision. While this is not absolutely necessary, it might be a valuable addition to have a brief discussion of this topic somewhere within the article. However, if the authors think otherwise, please ignore this comment.

3. Related to my comment 3, I think the concept of primary and secondary partners can be brought into the discussion. As polyamorous participants probably responded in relation to their primary partners (their longest relationship), it might result in some specific patterns of results that might differ if they had answered with another relationship in mind. See, e.g., https://doi.org/10.1371/journal.pone.0177841, https://doi.org/10.1007/s10508-018-1286-4, https://doi.org/10.1007/s10508-015-0658-2, as examples of differences between behaviors toward primary vs. secondary partners. I think if the authors discussed this issue, it would be a valuable addition to the article.

4. The authors did not make clear distinction between individuals in open relationships and people involved in swinging-type relationships. They were probably clustered together in one group. Though it is now not possible to divide this group into two, and I also think it would not be necessary, I think that a mere mention of such clustering may be added.

5. Related to the sentence, “(…) our results suggest that wellbeing cannot be solely explained by the simple distinction between monogamous and nonmonogamous relationships, but rather should consider other factors that go beyond relationship agreement”, I think authors may find interesting the idea of compersion, which is a common topic related to non-monogamy, especially polyamory. Some recent studies suggest that it may be an important factor in increasing relationship satisfaction in CNM communities. See: https://doi.org/10.1007/978-3-031-08956-5_2472-1, https://doi.org/10.1007/s10508-022-02333-4, https://doi.org/10.21203/rs.3.rs-2838247/v1 (full disclaimer: I am one of the authors of one of these works). The Authors may decide if this topic is suitable for mentioning in their Discussion section.

Despite minor issues mentioned above, I think that in overall the article is suitable for publication, presents well-performed analyses and interesting and novel results. Below I present the checklist of PLOS ONE criteria that articles may satisfy before publication with my short opinion:

1. The study presents the results of original research. YES

2. Results reported have not been published elsewhere. YES

3. Experiments, statistics, and other analyses are performed to a high technical standard and are described in sufficient detail. YES

4. Conclusions are presented in an appropriate fashion and are supported by the data. YES

5. The article is presented in an intelligible fashion and is written in standard English. YES

6. The research meets all applicable standards for the ethics of experimentation and research integrity. YES

7. The article adheres to appropriate reporting guidelines and community standards for data availability. NO (THE ACCESS TO DATA IS ON REQUEST)

Reviewer #3: This manuscript contains a cross-sectional study utilizing questionnaire methods to assess the various types of monogamous arrangements and their correlates. The manuscript is well written and I recommend that it be considered for publication. One general hesitation is that I don’t see what we can learn from this paper that is of novel interest to the field. Most of this appears to be consistent with what we already knew about the variations in monogamous relationships among LGBTQ individuals. Furthermore, I am a bit concerned about the way the analyses are conducted and reported, and these shortcomings must be addressed before this manuscript can be deemed publishable.

The authors claimed that their results showed null effects for the emotional and social well-being measures across various monogamy arrangements. But the results are presented in a confusing way—in some cases because the exact p-values are not given. For instance, for social well-being the adjusted p-value is .09, which I interpret as non-significant, but the p-value for emotional well-being is <.005, which is not an exact value but I interpret as statistically significant, and the authors demarcate statistically significant coefficients in that row using asterisks. To then conclude that “no significant differences were found among the five classes in terms of emotional, psychological, and social well-being reported by the participants” seems confusing and contradictory.

My second objection to the analyses as written is that the authors reported an “adjusted” set of p-values in the final column which they report in terms of a simultaneous regression model with demographic factors included. I am not sure why the authors made this analytic decision and they do not provide any explanation or justification for this. Why include age in this “adjusted” analysis? Because the authors did not preregister their analyses, it is possible that they added on these extra variables into the regression model after running an initial set of analyses. I am also not sure whether this is the technically correct use of the term “adjusted” which typically refers to a family-wise error rate correction. All of this confusion must be remedied.

The authors should report a one way analysis of variance (ANOVA) on the 5 classes of monogamy variations for each outcome variable. If they want to report an additional duplicate set of ACNOVAs with the other variables (e.g., age) included, they can do this, but not without a compelling justification for why those variables should be included. The authors should clearly and consistently report statistically significant findings in their tables and in their text, with family-wise error rate corrections where appropriate.

Regarding gender, it may be, as the authors contend, that lesbian women gravitate toward monogamy more than gay men because of “societal censure” that women face, and their true inclinations would be toward increased promiscuity, but this is being suppressed. This is certainly possible. However, another possibility that the authors neglect to mention is that there is a biological underpinning that motivates men towards greater sexual promiscuity compared to women, and this is true across sexual orientations. Heterosexual and gay men each have more positive attitudes toward promiscuous sexual behaviors compared to heterosexual and lesbian women, and the former also score higher on sociosexual orientation than the latter. In addition, the two citations following the authors’ social constructionist explanation do not appear to show evidence in support of this claim. One is an encyclopedia entry and the other is a study about attitudes toward monogamy in which the authors claim that “lesbians give more importance to monogamy but show less interest in starting a long-term relationship.” In any case, the authors’ preferred explanation may be correct but they should at least mention the alternative explanation of a biologically mediated difference between men and women in their sexual expression.

Minor point: I don’t understand why the authors lumped all non-white racial categories together into a BIPOC category. Black, Hispanic, Asian, and Native American populations belong to separate social and ethnic categories.

6. PLOS authors have the option to publish the peer review history of their article (what does this mean?). If published, this will include your full peer review and any attached files.

Reviewer #1: No

Reviewer #2: **Yes: **Klara Austeja Buczel

Reviewer #3: No

---

## [Author Response · Author response to Decision Letter 0]

12 Jul 2024

Below are the reviewers’ comment, followed by our responses. All changes in the manuscript are in red for your convenience.

REVIEWER #1: The literature cited is sufficiently extensive, however I would add some other recent contributions for the study of CNM such as the special section in the journal Archives of Sexual Behavior (Hamilton et al., 2021) and the special issue focusing on parenting practices published for Sexualities (Klesse et al., 2022). In reference to the intersection between plurisexual orientations and non-monogamies, other studies have highlighted how the exploration of plurisexuality often runs parallel with the exploration of consensual non-monogamy, or the exploration of non-monogamy is an incentive for the exploration of plurisexuality (and not just viceversa) (see Braida, 2021 in E. Maliepaard & R. Baumgartner (eds.), Bisexuality in Europe).

ANSWER: Thank you for your valuable feedback. We appreciate your suggestions for additional literature, and we have thoroughly reviewed the contributions you mentioned. In our revised manuscript, we have included the suggested references.

REVIEWER #1: The manuscript is well organized and written clearly enough to be accessible to non-specialists. I only recommend reviewing the use of parentheses, because several times in the text they are opened and not closed, or there are double parentheses.

ANSWER: Thank you for bringing this to our attention. We have addressed this issue.

REVIEWER #2: The authors conducted the study on a sample of LGBTQ+ people, but only on the cisgender group. Information about excluding transgender individuals (N = 190) and transferring their sample to another study is provided, but in my opinion, this step requires justification. At no point do the authors argue for reasons for such action, for example, by citing literature that shows that transgender individuals differ significantly from cisgender individuals in terms of relationships characteristics and other variables included in the study (if such literature exists). It would probably be valuable to also include this sample in this study and compare the results to those of the cisgender group. Otherwise, more justification of the exclusion of transgender individuals should be made.

ANSWER: After careful consideration, we agree that there is no reason for transgender individuals to be analyzed separately from the LGBQ sample. Therefore, we have pooled all participants’ data and updated the paper accordingly.

REVIEWER #2: Authors stated that “Polyamorous individuals were instructed to respond with their longest relationship in mind”. It is a common problem, namely, how to conduct research on CNM people to measure across all their partners. It may be problematic not only due to the complexity of such measurement, but also because biased responses by CNM individuals who could respond according to the anchoring heuristic or average their ratings, which would not provide adequate results. This comment is in no way intended to be a criticism of the authors, and I believe they made the right decision. While this is not absolutely necessary, it might be a valuable addition to have a brief discussion of this topic somewhere within the article. However, if the authors think otherwise, please ignore this comment.

Related to my comment 3, I think the concept of primary and secondary partners can be brought into the discussion. As polyamorous participants probably responded in relation to their primary partners (their longest relationship), it might result in some specific patterns of results that might differ if they had answered with another relationship in mind. See, e.g., https://doi.org/10.1371/journal.pone.0177841, https://doi.org/10.1007/s10508-018-1286-4, https://doi.org/10.1007/s10508-015-0658-2, as examples of differences between behaviors toward primary vs. secondary partners. I think if the authors discussed this issue, it would be a valuable addition to the article.

ANSWER: We agree with the Reviewer that measuring responses across all partners in CNM relationships presents significant challenges and potential biases. We appreciate the suggestion and have included a brief discussion about this issue in the manuscript to address the complexities and rationale behind our decision to focus on participants’ longest relationship. We now state on page 10:

“This decision was made to simplify the data collection and minimize response time. Yet, it may overlook the uniqueness of concurrent relationships, such as differences in investment, satisfaction, commitment, and communication between primary and secondary partners, which are critical to understanding the full spectrum of polyamorous experiences and relationships (Balzarini et al., 2017; Mogilski et al., 2015; Mogilski et al., 2019).”

REVIEWER #2: The authors did not make clear distinction between individuals in open relationships and people involved in swinging-type relationships. They were probably clustered together in one group. Though it is now not possible to divide this group into two, and I also think it would not be necessary, I think that a mere mention of such clustering may be added.

ANSWER: While we are able to distinguish between affective and sexual (non)exclusivity, our measure doesn’t allow us to make additional distinctions within these two overarching relationship types. Hence, specific identities and practices within sexually non-exclusive relationships (e.g., open vs swinging) remain masked. We have enhanced the variable description as well as the limitation of our measurement approach in the relevant sections of the paper.

REVIEWER #2: Related to the sentence, “(…) our results suggest that wellbeing cannot be solely explained by the simple distinction between monogamous and nonmonogamous relationships, but rather should consider other factors that go beyond relationship agreement”, I think authors may find interesting the idea of compersion, which is a common topic related to non-monogamy, especially polyamory. Some recent studies suggest that it may be an important factor in increasing relationship satisfaction in CNM communities. See: https://doi.org/10.1007/978-3-031-08956-5_2472-1, https://doi.org/10.1007/s10508-022-02333-4, https://doi.org/10.21203/rs.3.rs-2838247/v1 (full disclaimer: I am one of the authors of one of these works). The Authors may decide if this topic is suitable for mentioning in their Discussion section.

ANSWER: We thank the Reviewer for this insightful comment. We have decided to develop the section following the above suggestions. We now state on page 26:

“Some studies also suggest that compersion—the positive attitudes, thoughts, and actions that arise in response to one’s intimate partner’s extradyadic intimate relationships (Thouin-Savard et al., 2023)—significantly correlates with relationship satisfaction in consensually non-monogamous relationships (Thouin-Savard et al., 2023; Buczel et al., 2023; Flicker et al., 2022).”

REVIEWER #3: The authors claimed that their results showed null effects for the emotional and social well-being measures across various monogamy arrangements. But the results are presented in a confusing way—in some cases because the exact p-values are not given. For instance, for social well-being the adjusted p-value is .09, which I interpret as non-significant, but the p-value for emotional well-being is <.005, which is not an exact value but I interpret as statistically significant, and the authors demarcate statistically significant coefficients in that row using asterisks. To then conclude that “no significant differences were found among the five classes in terms of emotional, psychological, and social well-being reported by the participants” seems confusing and contradictory.

ANSWER: Thank you for bringing this to our attention. We apologize for any confusion caused by the presentation of our results. The omnibus p-value for the adjusted model was significant for emotional well-being (p < .005); however, after applying the Bonferroni correction, no significant between-group post-hoc differences were observed. We have rewritten the results section and revised the tables to improve readability and clarity. 

REVIEWER #3: My second objection to the analyses as written is that the authors reported an “adjusted” set of p-values in the final column which they report in terms of a simultaneous regression model with demographic factors included. I am not sure why the authors made this analytic decision and they do not provide any explanation or justification for this. Why include age in this “adjusted” analysis? Because the authors did not preregister their analyses, it is possible that they added on these extra variables into the regression model after running an initial set of analyses. I am also not sure whether this is the technically correct use of the term “adjusted” which typically refers to a family-wise error rate correction. All of this confusion must be remedied.

ANSWER: Regarding the inclusion of age in the adjusted analysis, this decision was made based on preliminary analyses identifying age as a confounder (i.e., significant correlations between age and the other variables under investigation). We acknowledge the importance of transparency in reporting analytic decisions and have provided a detailed rationale for the inclusion of age in the adjusted analysis in the revised manuscript.

REVIEWER #3: The authors should report a one way analysis of variance (ANOVA) on the 5 classes of monogamy variations for each outcome variable. If they want to report an additional duplicate set of ANCOVAs with the other variables (e.g., age) included, they can do this, but not without a compelling justification for why those variables should be included. The authors should clearly and consistently report statistically significant findings in their tables and in their text, with family-wise error rate corrections where appropriate.

ANSWER: While Latent Class Analysis technically doesn’t allow for direct AN(C)OVA, as post-hoc analyses must take into account the posterior probabilities of class memberships to produce valid estimates, we conducted a step 3 “ANOVA-like” analysis on distal outcomes to examine differences in outcomes across classes (Vermunt & Magidson, 2013). We have changed the way we present these results in Table 5. We now reporte bivariate results using >, <, and = symbols, and use bold characters to emphasize between-class differences that remained significant after applying the Bonferroni family-wise error rate correction. We believe that this new approach makes the results clearer, and the statistical modeling choices, more transparent.

Vermunt, J. K., & Magidson, J. (2013). Technical guide for Latent GOLD 5.0: Basic, advanced, and syntax. Belmont, MA: Statistical Innovations Inc.

REVIEWER #3: Regarding gender, it may be, as the authors contend, that lesbian women gravitate toward monogamy more than gay men because of “societal censure” that women face, and their true inclinations would be toward increased promiscuity, but this is being suppressed. This is certainly possible. However, another possibility that the authors neglect to mention is that there is a biological underpinning that motivates men towards greater sexual promiscuity compared to women, and this is true across sexual orientations. Heterosexual and gay men each have more positive attitudes toward promiscuous sexual behaviors compared to heterosexual and lesbian women, and the former also score higher on sociosexual orientation than the latter. In addition, the two citations following the authors’ social constructionist explanation do not appear to show evidence in support of this claim. One is an encyclopedia entry and the other is a study about attitudes toward monogamy in which the authors claim that “lesbians give more importance to monogamy but show less interest in starting a long-term relationship.” In any case, the authors’ preferred explanation may be correct but they should at least mention the alternative explanation of a biologically mediated difference between men and women in their sexual expression.

ANSWER: We thank the Reviewer for this comment. We have updated our discussion to include acknowledgment of the alternative explanation of biological factors influencing gender differences in sexual behavior. We now state on page 24:

Some evolutionary psychology hypotheses suggest the existence of sex differences in mating strategies, with cisgender women being more likely to endorse monogamy, and cisgender men, non-monogamy (82,83). 

REVIEWER #3: I don’t understand why the authors lumped all non-white racial categories together into a BIPOC category. Black, Hispanic, Asian, and Native American populations belong to separate social and ethnic categories.

ANSWER: We agree with the Reviewer’s observation regarding the grouping of non-white racial categories into a single BIPOC category. In response to this concern, we have decided to remove this variable from our analyses. The manuscript has been updated accordingly.

---

## [Decision Letter · Decision Letter 1]

22 Aug 2024

Romantic relationship configurations and their correlates among LGBTQ+ persons: A latent class analysis

PONE-D-24-03060R1

Dear Dr. Blais,

We’re pleased to inform you that your manuscript has been judged scientifically suitable for publication and will be formally accepted for publication once it meets all outstanding technical requirements.

Kind regards,

Ning Cai, Ph.D.

Section Editor

PLOS ONE

Additional Editor Comments (optional):

Please refer to the reviewers' comments when preparing the final version.

Reviewers' comments:

Reviewer's Responses to Questions

**Comments to the Author**

1. If the authors have adequately addressed your comments raised in a previous round of review and you feel that this manuscript is now acceptable for publication, you may indicate that here to bypass the “Comments to the Author” section, enter your conflict of interest statement in the “Confidential to Editor” section, and submit your "Accept" recommendation.

Reviewer #1: All comments have been addressed

Reviewer #2: All comments have been addressed

2. Is the manuscript technically sound, and do the data support the conclusions?

Reviewer #1: Yes

Reviewer #2: Yes

3. Has the statistical analysis been performed appropriately and rigorously? 

Reviewer #1: Yes

Reviewer #2: Yes

4. Have the authors made all data underlying the findings in their manuscript fully available?

Reviewer #1: Yes

Reviewer #2: No

5. Is the manuscript presented in an intelligible fashion and written in standard English?

Reviewer #1: Yes

Reviewer #2: Yes

6. Review Comments to the Author

Reviewer #1: (No Response)

Reviewer #2: The authors responded sufficiently to all my comments. I recommend the manuscript for publication.

I have one very small comment as the last one - literature item No. 91 (Buczel et al., 2023) refers to the preprint, while the article was officially published this year. Authors may change this citation according to this one below:

91. Buczel KA, Szyszka PD, Mara I. Exploring Compersion: A Study on Polish Consensually

Non-Monogamous Individuals and Adaptation of the COMPERSe Questionnaire. Arch Sex Behav. 2024 Jul 1. https://doi.org/10.1007/s10508-024-02930-5

7. PLOS authors have the option to publish the peer review history of their article (what does this mean?). If published, this will include your full peer review and any attached files.

Reviewer #1: No

Reviewer #2: No

---

## [Editor Report · Acceptance letter]

4 Sep 2024

PONE-D-24-03060R1 

PLOS ONE

Dear Dr. Blais, 

I'm pleased to inform you that your manuscript has been deemed suitable for publication in PLOS ONE. Congratulations! Your manuscript is now being handed over to our production team.

Kind regards, 

on behalf of

Dr. Ning Cai 

Section Editor

PLOS ONE